# Musashi-1 and miR-147 Precursor Interaction Mediates Synergistic Oncogenicity Induced by Co-Infection of Two Avian Retroviruses

**DOI:** 10.3390/cells11203312

**Published:** 2022-10-21

**Authors:** Defang Zhou, Longying Ding, Menglu Xu, Xiaoyao Liu, Jingwen Xue, Xinyue Zhang, Xusheng Du, Jing Zhou, Xiyao Cui, Ziqiang Cheng

**Affiliations:** College of Veterinary Medicine, Shandong Agricultural University, Tai’an 271018, China

**Keywords:** ALV-J, REV, Musashi-1, miR-147 precursor, synergistic tumorigenesis, co-infection, NF-κB/KIAA1199/EGFR pathway

## Abstract

Synergism between avian leukosis virus subgroup J (ALV-J) and reticuloendotheliosis virus (REV) has been reported frequently in co-infected chicken flocks. Although significant progress has been made in understanding the tumorigenesis mechanisms of ALV and REV, how these two simple oncogenic retroviruses induce synergistic oncogenicity remains unclear. In this study, we found that ALV-J and REV synergistically promoted mutual replication, suppressed cellular senescence, and activated epithelial-mesenchymal transition (EMT) in vitro. Mechanistically, structural proteins from ALV-J and REV synergistically activated the expression of Musashi-1(MSI1), which directly targeted pri-miR-147 through its RNA binding site. This inhibited the maturation of miR-147, which relieved the inhibition of NF-κB/KIAA1199/EGFR signaling, thereby suppressing cellular senescence and activating EMT. We revealed a synergistic oncogenicity mechanism induced by ALV-J and REV in vitro. The elucidation of the synergistic oncogenicity of these two simple retroviruses could help in understanding the mechanism of tumorigenesis in ALV-J and REV co-infection and help identify promising molecular targets and key obstacles for the joint control of ALV-J and REV and the development of clinical technologies.

## 1. Introduction

Avian leukosis virus subgroup J (ALV-J), the sixth subgroup of the *Alpharetrovirus* genus in the family Retroviridae, was identified in meat-type breeder chickens in 1991 [1,2]. Reticuloendotheliosis virus (REV), a gammaretrovirus of the family Retroviridae, was identified in adult turkeys in 1958 [3,4]. ALV-J and REV are transmitted both horizontally and vertically [5,6], and are the most common naturally occurring simple retroviruses associated with neoplastic and immunosuppressive diseases in poultry [7,8,9,10]. Owing to the similar characteristics and widespread nature of ALV-J and REV in the field, co-infection is common [11,12,13,14,15,16] and is an important emerging problem in chicken flocks. Synergism in ALV-J and REV co-infected chicken flocks causes higher mortality, more serious growth retardation, and immunosuppression [17,18], and facilitates viral replication and exosomal miRNA accumulation [19]. Synergism is a common phenomenon in retroviruses [20,21,22,23].

ALV-J induces late-onset myelocytomas, hemangiomas, and various other tumors in chickens, and REV induces chronic lymphomas in chickens, ducks, geese, pheasants, quails, and turkeys [24]. The two avian retroviruses continue to be of great interest in understanding the molecular mechanisms of tumorigenesis. In ALV-J and REV, the induction of neoplasms occurs in a minority of cases and only after several months of infection, presenting great difficulties in exploring the mechanisms of synergistic tumorigenesis. Since the suppression of cellular senescence and the activation of EMT are prerequisites for neoplasm and metastasis, many studies have considered cellular senescence and EMT as essential indicators of cellular oncogenicity in vitro [25,26,27,28,29,30].

Co-infection with oncogenic viruses has been increasingly reported in recent years [31,32]. Co-infection plays an important role in the development of neoplasms. The two co-infected oncogenic viruses can assist each other during the initiative and developmental process of the tumor. For retrovirus, slow (cis-activation) and acute (trans-activation) transformation are two classical oncogenicity mechanisms [6]. Slowly transforming retroviruses are considered replication-competent and do not carry oncogenes [33] that induce late-onset tumors through insertional mutagenesis by activating cellular proto-oncogenes or inactivating tumor suppressor genes [34,35,36,37,38]. If a slowly transforming retrovirus obtains a proto-oncogene from the host genome, it becomes an acutely transforming virus that can induce rapid-onset tumors within a short time after host infection, such as myc in MC29, CMII and OK10 strains of ALV [6], erbB in avian erythroblastosis virus [39], and v-rel in the T-strain of REV [40]. However, it has not been fully elucidated whether the synergism of the two oncogenic viruses further affects the interaction between the viruses and host cells or can further modify signaling pathways.

In the present study, we revealed a synergistic oncogenicity mechanism induced by two simple oncogenic retroviruses, ALV-J and REV, which can help to identify promising molecular targets and key obstacles for the joint control of ALV-J and REV and in the development of clinical technology.

## 2. Materials and Methods

### 2.1. Cells, Viruses, and Plasmids

Primary chicken embryo fibroblasts (CEFs), DF-1 cells (a spontaneously immortalized CEF cell line), and human 293T cells, maintained in the laboratory of animal pathology of Shandong Agriculture University, were cultured in Dulbecco’s modified Eagle’s medium (DMEM) supplemented with 10% fetal bovine serum (FBS), 1% penicillin/streptomycin, and 1% l-glutamine, and incubated at 37 °C in a 5% CO_2_ incubator. The stock SNV strain of REV at 10^3.2^ 50% tissue culture infectious dose (TCID_50_) and the NX0101 strain of ALV-J at 10^3.8^ TCID_50_ were maintained in the laboratory of animal pathology of Shandong Agriculture University. The TCID_50_ of the SNV and NX0101 strains was titrated by limiting dilution in the DF-1 culture. KIAA1199 and NF-κB p50 3′UTRs were cloned downstream of the luciferase reporter gene of the pmirGLO control vector to create wild-type pmirGLO-KIAA1199 3′UTR (WT KIAA1199) and pmirGLO-p50 3′UTR (WT p50) plasmids, respectively (GenePharma, Shanghai, China). Bioinformatics analysis software tools and websites, including RNA22, NCBI, ENSEMBLE, and the RNA-protein interaction prediction website (http://pridb.gdcb.iastate.edu/RPISeq/, accessed on 28 September 2022), were used to analyze and predict the binding sites between miR-147 and the 3′ UTR regions of KIAA1199 or NF-κB p50. The pmirGLO-KIAA1199 3′ UTR mutant plasmid (Mut KIAA1199) and the pmirGLO-p50 3′ UTR mutant plasmid (Mut p50) were constructed through site-directed mutagenesis. miR-147 mimics, miR-147 inhibitors, Flag-MSI1, MSI1 Cas9/gRNA, Flag-KIAA1199, KIAA1199 Cas9/gRNA, KIAA1199 shRNAs, Flag-NF-κB p50, NF-κB p50 Cas9/gRNA, NF-κB p50 shRNAs, Flag-ALV-J gag, Flag-ALV-J pol, Flag-ALV-J env, HA-REV gag, HA-REV pol, and HA-REV env plasmids were purchased from GenePharma (Shanghai, China). The construction of MSI1 mutant plasmids was performed using the Fast Site-Directed Mutagenesis Kit (TIANGEN, Beijing, China) according to the manufacturer’s instructions.

### 2.2. Establishment of a Tumor Model and Sample Collection in Specific Pathogen-Free Chickens

This study was performed in strict accordance with the recommendations of the Shandong Institutional Animal Care and Use Committee. Ethical approval for this study was obtained from the Ethics Committee of Animal Experiments in Shandong Province (permit no. SDAU 15-124). Specific pathogen-free chicken embryos (120 embryos) were purchased from SPAFAS Co. (Jinan, China; a joint venture with Charles River Laboratory, Wilmington, MA, USA), allocated into four groups, and placed in separate incubators supplied with filtered positive-pressure air. Thirty embryos were each inoculated with ALV-J (100 μL, 10^3.8^ TCID_50_) and 100 μL DMEM per egg, or REV (100 μL, 10^3.2^ TCID_50_) and 100 μL DMEM per egg, or both ALV-J (100 μL, 10^3.8^ TCID_50_) and REV (100 μL, 10^3.2^ TCID_50_) per egg, through the allantoic cavity at an embryonic age of 6 days. Mock-infected embryos were inoculated with 200 μL DMEM. Of these embryos, 15, 19, and 10 hatched in the ALV-J (15/30; 50.0%), REV (19/30; 63.3%), and co-infection groups (10/30; 33.3%), respectively. In comparison, 90% (27/30) of the uninoculated control chickens hatched. The chickens were observed daily and euthanized when they were apparently ill or at 18 weeks of age. Of the 44 virus-infected chickens, 13 died at weeks 3, 4, and 10 for reasons unrelated to the infection. Ten chickens in the ALV-J group, 13 chickens in the REV group, and 8 chickens in the co-infection group were euthanized at 18 weeks of age, and 4 of 10, 0 of 13, and 7 of 8 bore tumors. In total, tumor tissues were obtained from 4 ALV-infected and 7 co-infected chickens. Myelocytomas, fibromas, lymphomas, and corresponding non-cancerous tissues, including the bone marrow, liver, and heart from each chicken, were divided into three portions for Western blotting (WB), ELISA, and RNA extraction assays.

### 2.3. Hematoxylin and Eosin Staining

The cancerous tissues and corresponding non-cancerous tissues of chickens were formalin-fixed, paraffin-embedded, sectioned, and stained for histopathological observation.

### 2.4. Illumina Small-RNA Deep Sequencing

Total RNA from infected CEF cell samples, either mock-infected or infected with ALV-J or REV alone, or co-infected with both ALV-J and REV for 72 hpi, was separated on 15% agarose gels to extract small RNA (18–30 nt). Illumina small-RNA deep sequencing was performed as previously described [19].

### 2.5. TMT-Labeled LC−MS/MS

SDT buffer was added to the CEF cell samples, and the same batch of samples was used for Illumina small RNA deep sequencing. The lysate was sonicated and boiled for 15 min. After centrifugation at 14,000× *g* for 40 min, the supernatant was quantified using a BCA Protein Assay Kit (Bio-Rad, Hercules, CA, USA). Proteins from each sample were separated using SDS-PAGE, prepared using a sample filter-aid, and labeled according to the manufacturer’s instructions (Applied Biosystems, Carlsbad, CA, USA). The TMT-labeled peptides were fractionated by SCX chromatography using an AKTA Purifier system (GE Healthcare, Chicago, IL, USA). Each fraction was subjected to nanoLC-MS/MS analysis. LC-MS/MS analysis was performed using a Q Exactive mass spectrometer (Thermo Fisher Scientific, Waltham, MA, USA). MS/MS spectra were searched using the MASCOT engine (version 2.2; Matrix Science, London, UK) embedded in the Proteome Discoverer 1.4.

### 2.6. Luciferase Reporter Assays

The miRNA mimic, KIAA1199 3 UTR, or NF-κB p50 3′ UTR luciferase reporter plasmids were co-transfected into CEF cells. At 48 hpi, cell lysates were prepared according to the manufacturer’s instructions using the Dual-Lumi™ Luciferase Reporter Gene Assay Kit (Beyotime Co., Ltd., Shanghai, China). Luciferase activity was measured using the dual-luciferase reporter assay system (Beyotime Co., Ltd.) and normalized against the activity of the Renilla luciferase gene.

### 2.7. Western Blotting

The cells were lysed using cell lysis buffer (Beyotime) and incubated on ice for 5 min. Lysates were resuspended in SDS loading buffer, boiled for 5 min, loaded, and run on a 10% SDS-PAGE gel, and then transferred onto a nitrocellulose membrane (Solarbio, Beijing, China). Membranes were blocked with 5% skimmed milk at 4 °C overnight and probed with anti-ALV-J env (mouse monoclonal W459, Animal Pathology Lab, Shandong Agriculture University, Taian, China), anti-REV env (mouse monoclonal W460, Animal Pathology Lab, Shandong Agriculture University), anti-N-cadherin (rabbit monoclonal ab76011, Abcam, Cambridge, UK), anti-E-cadherin (rabbit monoclonal ab40772, Abcam), anti-vimentin (rabbit monoclonal ab92547, Abcam), anti-SNAIL (rabbit polyclonal ab85936, Abcam), anti-Flag (mouse monoclonal AT0022, Engibody, DE, USA), anti-HA (mouse monoclonal AT0024, Engibody), anti-KIAA1199 (rabbit polyclonal bs-21528R, Bioss, Beijing, China), anti-EGFR (rabbit polyclonal bs-10007R, Bioss), anti-MSI1 (rabbit monoclonal ab52865, Abcam), and anti-p50 (rabbit polyclonal bs-1194R, Bioss) antibodies at 1:1000, 1:1000, 1:1000, 1:1000, 1:1000, 1:1000, 1:1000, 1:1000, 1:200, 1:400, 1:2000, and 1:1000 dilutions, respectively, followed by horseradish peroxidase (HRP)-conjugated goat anti-rabbit secondary antibody (Engibody) or HRP-conjugated goat anti-mouse secondary antibody (Engibody) at a dilution of 1:3000. β-actin was used as the loading control. Protein levels were detected using the Enhanced HRP-DAB Chromogenic Substrate Kit (Tiangen), according to the manufacturer’s instructions.

### 2.8. Quantitative Real-Time Polymerase Chain Reaction

The specific primer sequences for pri-miR-147, KIAA1199, NF-κB p65, NF-κB p50, MSI1, EGFR, and GAPDH used in this study are listed in Appendix A. Total RNA from CEF cells that had been either mock-infected, mono-infected with ALV-J or REV, or co-infected with ALV-J and REV was isolated using the Tiangen RNeasy mini kit (TIANGEN) according to the manufacturer’s instructions, with optional on-column DNase digestion. RNA integrity and concentrations were assessed by means of agarose gel electrophoresis and spectrophotometry, respectively. RNA (1 µg per triplicate reaction) was reverse-transcribed to cDNA using the Taqman Gold Reverse Transcription kit (Applied Biosystems). Real-time RT-PCR (qRT-PCR) was performed using SYBR^®^ Premix Ex Taq, and specific primers (Appendix A). All values were normalized to endogenous GAPDH levels to control for variation. For qRT-PCR analysis of miR-147, we used an miRcute miRNA first-stand cDNA synthesis kit and an miRcute miRNA qPCR detection kit (SYBR Green) (TIANGEN). The reverse primer provided in the miRcute miRNA qPCR detection kit was complementary to the poly (T) adapter. Data were collected on an ABI PRISM 7500 and analyzed using Sequence Detector v1.1 software (Applied Biosystems, USA). All values were normalized to endogenous U6 to control for variations. The primers specific for U6 are listed in Appendix A. Assays were performed in triplicate, after which the average threshold cycle (CT) values were used to determine relative concentration differences based on the ΔΔCT method of relative quantization described in the manufacturer’s protocol.

### 2.9. Determination of NF-κB p65 Nuclear Translocation

The translocation of NF-κB p65 from the cytoplasm to the nucleus was examined via immunofluorescence. CEFs were washed with PBS and fixed with 4% paraformaldehyde for 20 min. After fixation, the cells were permeabilized with 0.25% Triton X-100 in PBS for 10 min and blocked with 10% BSA in PBS for 1 h. Anti-NF-κB p65 antibody (1:100, Bioss) was incubated overnight at 4 °C, followed the next day by a one-hour incubation at room temperature with anti-rabbit IgG antibody labeled with FITC (1:1000, Engibody). Finally, the cells were washed with PBS, incubated with DAPI for 5 min, and then observed under a fluorescence microscope after washing with PBS.

### 2.10. Senescence-Associated β-Gal Staining

A senescence β-galactosidase staining kit was purchased from Beyotime (Shanghai, China) and was used to evaluate cellular senescence according to the manufacturer’s instructions.

### 2.11. ELISA for NF-κB p65 and P-IκBα Assays

Chicken NF-κB p65 and chicken NF-κB P-IκBα ELISA kits were purchased from Senbeijia (Nanjing, China) and were used to determine the expression levels of NF-κB p65 according to the manufacturer’s instructions.

### 2.12. RNA ChIP Assay

The RNA ChIP kit was purchased from Active Motif (Shanghai, China) and was used to assay RNA–protein interactions according to the manufacturer’s instructions.

### 2.13. Statistical Analysis

Data are presented as the mean ± standard deviation(s). The *t*-test and one-way ANOVA tests were performed using SPSS v. 13.0 statistical software (SPSS, Chicago, IL, USA). Statistical significance was set at *p* ≤ 0.05.

## 3. Results

### 3.1. ALV-J and REV Synergistically Suppress Cellular Senescence and Activate Epithelial–Mesenchymal Transition (EMT) In Vitro

The suppression of cellular senescence or activation of EMT is a prerequisite for tumorigenesis and metastasis. To understand whether ALV-J and REV synergistically affected oncogenicity, cellular senescence and EMT were measured in ALV-J and REV co-infected cells. The RNA levels and protein levels of both ALV-J and REV were increased significantly in co-infected cells compared to those in mono-infected cells (Figure 1A,B,E–G), confirming that there was synergistic replication between ALV-J and REV. Senescence-associated (SA)-β-Gal staining revealed that ALV-J and REV synergistically inhibited cellular senescence (Figure 1C,D). The expression levels of EMT-associated proteins, assessed via Western blotting (WB) analysis, suggested that ALV-J and REV synergistically activated the EMT process (Figure 1E–G). These findings confirm that ALV-J and REV synergistically suppress cellular senescence and activate the EMT process in vitro.

### 3.2. Identification of Key Host Molecules Responsible for Synergistic Oncogenicity Induced by ALV-J and REV

Chick embryo fibroblasts (CEFs) co-infected with ALV-J and REV, mono-infected with ALV-J or REV, and mocks were analyzed using tandem mass tag (TMT)-based proteomics combined with miRNA whole-genome sequencing analysis. Among the 33 differentially expressed proteins and 17 differentially expressed miRNAs (Figure 2A,B), only miR-147 (or pri-miR-147) exhibited potential interactions with Musashi-1 (MSI1), NF-κB p50, and KIAA1199, which are associated with the cancer signaling pathway. The decreased miR-147 expression was verified using qPCR (Figure 2C), and the increased expression of MSI1, KIAA1199, and NF-κB p50 were verified using WB in co-infected CEFs (Figure 2D). RNA–protein interaction analysis showed interactions between miR-147 and MSI1, KIAA1199, and NF-κB p50 in ALV-J and REV co-infected cells. These results suggest that the interaction between miR-147 and activated MSI1, KIAA1199, and NF-κB p50 may be important in the synergistic oncogenicity induced by ALV-J and REV.

### 3.3. Ectopic Expression of miR-147, MSI1, KIAA1199, and NF-κB p50 Is Associated with Oncogenicity

To verify the association of miR-147, MSI1, KIAA1199, and NF-κB p50 with oncogenicity, we measured cellular senescence and EMT through the construction and transfection of miR-147 mimics, miR-147 inhibitors, FLAG-MSI1, MSI1 Cas9/gRNA, FLAG-KIAA1199, KIAA1199 Cas9/gRNA, KIAA1199 shRNAs, FLAG-NF-κB p50, NF-κB p50 Cas9/gRNA, and NF-κB p50 shRNAs in CEF cells. SA-β-Gal staining and WB assays revealed that the miR-147 inhibitor MSI1, KIAA1199, or NF-κB p50 blocked cellular senescence and promoted EMT (Figure 3A–D), whereas miR-147 mimics, MSI1 knockdown, KIAA1199 knockdown, and NF-κB p50 knockdown promoted cellular senescence and inhibited EMT (Figure 3E–H). These data suggest that miR-147, MSI1, KIAA1199, and NF-κB p50 are associated with oncogenicity.

### 3.4. MSI1 Directly Targeted pri-miR-147 to Inhibit miR-147 Maturation

We intended to detect the expression of the miR-147 precursor (pri-miR-147) to determine whether mature miR-147 was inhibited before or after pri-miR-147 transcription. In contrast to mature miR-147, ALV-J and REV synergistically activated the expression of pri-miR-147 rather than inhibiting it. This showed that miR-147 was inhibited after pri-mir147 transcription (Figure 4A). The relationship between MSI1 and pri-miR-147/miR-147 was further analyzed. MiR-147 levels were detected when MSI1 was over- or under-expressed in CEFs. When MSI1 was overexpressed, the levels of mature miR-147 were suppressed more than 3.77-fold (Figure 4B). Upon MSI1 knockdown, the levels of mature miR-147 were elevated more than 5.58-fold (Figure 4C). Furthermore, RNA chromatin immunoprecipitation (ChIP) revealed over 15.6-fold enrichment of pri-miR-147, which is associated with MSI1 (Figure 4D,E), indicating that MSI1 directly targeted pri-miR-147 RNA. To identify the domain in MSI1 that binds pri-miR-147, we constructed four MSI1 mutants based on its RNA-binding sites [41,42], and transfected CEFs to detect mature miR-147 levels. The four MSI1 mutants are shown in Figure 4F. WB analysis confirmed that all MSI mutants were successfully transfected into the CEF cell line DF-1 (Figure 4G). The miR-147 expression level showed that only MSI1 mut1 relieved the inhibition of miR-147 maturation (Figure 4H), implying that the RNA-binding site (amino acid sequences 33 and 35 to 39) was the key domain for inhibiting miR-147 maturation. These data suggest that MSI1 directly targets pri-miR-147 through its RNA binding site (amino acid sequences 33 and 35 to 39), leading to the inhibition of miR-147 maturation.

### 3.5. miR-147 Targets NF-κB p50 and KIAA1199

To confirm that miR-147 directly targeted KIAA1199 and NF-κB p50, a dual-luciferase assay was performed in CEFs. The KIAA1199 3′ untranslated region (UTR) luciferase reporter assay revealed that miR-147 significantly inhibited the activity of the KIAA1199 3′ UTR reporter and that of the NF-κB p50 3′ UTR reporter, but not that of control reporters (Figure 5A,B). miR-147 inhibited the activities of the KIAA1199 3′ UTR reporter and the NF-κB p50 3′ UTR reporter and suppressed the endogenous expression levels of KIAA1199 and NF-κB p50 in CEF cells in a dose-dependent manner (Figure 5C–E). In contrast, the miR-147 inhibitor upregulated the expression of endogenous KIAA1199 and NF-κB p50 in a dose-dependent manner (Figure 5F). Bioinformatics analysis identified one putative miR-147 binding site at the KIAA1199 3′ UTR and the NF-κB p50 3ʹ UTR, respectively (Figure 4G,H). Mutations in the putative miR-147 binding site eliminated the inhibitory effect of miR-147 on the reporter activities of the KIAA1199 3′ UTR and NF-κB p50 3′ UTR (Figure 5I,J). These findings suggested that miR-147 directly targeted KIAA1199 and NF-κB p50.

### 3.6. ALV-J and REV Synergistically Activated the NF-κB/KIAA1199/EGFR Signaling Pathway

Previous studies have shown that KIAA1199, an oncogene that is transcriptionally induced by NF-κB proteins, promotes EGFR stability and contributes to the activation of NF-κB/EGFR signaling pathway crosstalk in breast cancer [43,44]. To determine whether EGFR is involved in NF-κB/KIAA1199 signaling that is synergistically activated by ALV-J and REV, expression levels of NF-κB p65, phosphorylated IκBα, and EGFR were detected using qPCR, ELISA, and WB. Compared to mono-infection, NF-κB p65 RNA levels in co-infected cells were elevated 1.78- and 1.91-fold, respectively (Figure 6A). ELISA was used to confirm these results (Figure 6B). Compared with single infection, translocation of NF-κB to the nucleus was significantly observed in CEF-co-infected ALV-J and REV (Appendix A). The increase in phosphorylated IκBα levels indicated that ALV-J and REV synergistically activated the NF-κB signaling pathway (Figure 6C). Furthermore, we found that ALV-J and REV synergistically enhanced EGFR expression levels (Figure 6D,E). To determine the correlation between NF-κB, KIAA1199, and EGFR in ALV-J and REV co-infected cells, RNA interference was carried out via the construction and transfection of NF-κB p65 or KIAA1199 shRNAs into DF-1 cells. Upon NF-κB p65 or KIAA1199 knockdown, the expression levels of KIAA1199 and EGFR (Figure 6F–H) or EGFR and NF-κB p65 were suppressed (Figure 6I–K). Taken together, these findings suggested that ALV-J and REV synergistically activate the NF-κB/KIAA1199/EGFR signaling pathway.

### 3.7. Structural Proteins, Especially Gags from ALV-J and REV, Synergistically Activate MSI1

The genome of simple retroviruses is composed of gag, pol, and env, which encode core proteins, proteases, and envelope proteins, respectively [45]. To determine whether the synergistic activation of MSI1 by ALV-J and REV is caused by its structural proteins, we detected the expression of MSI1 in cells infected with REV and transfected with ALV-J structural proteins (gag, pol, and env), or infected with ALV-J and transfected with REV structural proteins (gag, pol, and env). All viral structural proteins showed the synergistic promotion of MSI1 expression; however, ALV-J gag and REV gag showed the most significant promotion effect on MSI1 expression (Figure 7A,B). Furthermore, co-transfection of the two gags from ALV-J and REV demonstrated synergism in regard to MSI1 activation (Figure 7C). These findings confirmed that all structural proteins of ALV-J and REV were involved in the synergistic activation of MSI1.

### 3.8. MSI1-miR-147 Regulated NF-κB/KIAA1199/EGFR Pathways Present in Tumors Induced by ALV-J and REV

To verify the synergistic oncogenicity occurring in ALV-J and REV co-infected tumor-bearing chickens, we established a tumor model induced by ALV-J and REV and measured the key molecules and the NF-κB/KIAA1199/EGFR pathway in three type tumors. All chickens were euthanized at 18 weeks of age; 4 of 10 (40%) ALV-J infected chickens, 0 of 13 (0%) REV-infected chickens, and 7/8 (87.5%) co-infected chickens bore tumors. Histopathological examination showed that the tumors induced by ALV-J were myelocytomas; however, ALV-J and REV induced myelocytomas, lymphomas, and endocardial fibromas (Figure 8A). These findings suggested that ALV-J and REV synergistically promoted tumorigenesis in chickens. To validate the results of the in vitro experiments, we detected the RNA levels of miR-147, MSI1, KIAA1199, NF-κB p50, and EGFR in the cancerous tissues of chickens in different infection groups, including the bone marrow, liver, and heart. Compared to the mono-infection group, the RNA levels of miR-147 were significantly downregulated, whereas MSI1, KIAA1199, NF-κB p50, and EGFR were significantly upregulated in the bone marrow, livers, and hearts of chickens (Figure 8B–F). Next, we determined the RNA expression levels of miR-147 in five cases of myelocytomas, lymphomas, and endocardial fibromas. Compared to non-cancerous tissues in co-infected chickens, miR-147 levels in myelocytomas, fibromas, and lymphomas in four, five, and five cases, respectively, were suppressed (Figure 9A). Compared to non-cancerous tissues, ELISA revealed that four, five, and four cases of phosphorylated IκBα levels were elevated in myelocytomas, fibromas, and lymphomas, respectively (Figure 9B). Simultaneously, elevated expression levels of EGFR, KIAA1199, NF-κB p50, and MSI1 were also confirmed in myelocytomas, fibromas, and lymphomas using WB (Figure 9C–E). Compared to the corresponding non-cancerous tissues, MSI1 and NF-κB/KIAA1199/EGFR pathway crosstalk was upregulated in 15 of 15 and 13 of 15 tumors, respectively, whereas miR-147 was downregulated in 14 of 15 tumors (Figure 9F). These data suggest that a synergistic tumorigenesis mechanism occurred in chickens co-infected with ALV-J and REV.

## 4. Discussion

Synergistic interactions between two retroviruses in co-infected hosts have been well documented [20,21,22,23]. Co-infection with two or more oncogenic retroviruses is known to accelerate cancer development [32]. Recent studies have shown that co-infection of ALV-J and REV causes higher mortality, more serious growth retardation, and immunosuppression, facilitating viral replication and changing the miRNA expression profile [17,18,19]. However, the question of whether synergism promotes oncogenicity and the underlying synergistic mechanism remains unclear. In this study, we found that ALV-J and REV synergistically suppressed cellular senescence and activated EMT in vitro, indicating that these two viruses have developed strategies to synergistically promote oncogenic potential in vitro.

The suppression of cellular senescence and the activation of EMT are prerequisites for neoplasm and metastasis, which have been commonly considered as essential indicators of cellular oncogenic potential in vitro [25,26,27,28,29,30]. To identify the key molecules responsible for the synergistic oncogenicity induced by ALV-J and REV in host cells, TMT-based proteomics, combined with miRNA whole-genome sequencing, was used to screen and identify the key molecules in the co-infected/mono-infected/mock cells. Interestingly, an miRNA molecule, miR-147, known as a tumor suppressor [46,47,48,49] showed ectopic expression, which increased in mono-infected cells and decreased in co-infected cells. However, its precursor, pri-miR-147, showed a synergistic increase in ALV-J and REV-co-infected cells. These data suggested that a certain molecule blocks miR-147 maturation, releasing its target signals for tumorigenesis. Thus, target analysis indicated that miR-147 or pri-miR-147 exhibited potential interactions with MSI1, NF-κB p50, and KIAA1199, which are associated with the cancer signaling pathway. Next, we demonstrated that miR-147, MSI1, NF-κB p50, and KIAA1199 play a critical role in cellular senescence suppression and EMT activation, indicating that these molecules are involved in the synergistic oncogenicity induced by ALV-J and REV.

MSI1, an RNA-binding protein, has been found to regulate multiple critical biological processes that are relevant to cancer initiation and progression [50]. KIAA1199, a novel proto-oncogene, has been associated with tumor progression and metastasis in numerous cancers [51]. NF-κB activation is key to the early development of some cancers [52]. Based on the functions of these molecules, we speculated that MSI1 blocks the maturation of miR-147, which relieves the inhibition of NF-κB p50 and KIAA1199. The experimental results support our speculation. MSI1 directly targeted pri-miR-147 through its RNA-binding site, inhibiting miR-147 maturation. Because miR-147 directly targets NF-κB p50 and KIAA1199, downregulation of miR-147 promoted the upregulation of NF-κB p50 and KIAA1199. Recent studies have demonstrated that NF-κB and EGFR are partners in cancer, and NF-κB-induced KIAA1199 promotes EGFR stability, contributing to the activation of the NF-κB/EGFR signaling pathway [43,44]. Because EGFR was absent in the TMT-based proteomics results, we wanted to know whether EGFR is involved in the NF-κB/KIAA1199 signaling that is synergistically activated by ALV-J and REV. The results showed that EGFR participated in the NF-κB/KIAA1199 pathway, namely, NF-κB/KIAA1199/EGFR, which was synergistically activated by ALV-J and REV.

We observed that ALV-J and REV synergistically activate MSI1, which binds pri-miR-147, blocking miR-147 maturation and thereby relieving the inhibition of the NF-κB/KIAA1199/EGFR signaling pathway. We then investigated how ALV-J and REV synergistically activate MSI1. The results showed that all structural proteins, especially gags from ALV-J and REV, synergistically activated the expression of MSI1. Generally, gag proteins from complex retroviruses or acutely transforming retroviruses (carrying gag-onc fusion genes) are involved in oncogenesis [53,54]. Here, we observed for the first time that gags from two simple retroviruses synergistically promoted oncogenicity. Finally, the key molecules and signaling pathways involved in synergistic tumorigenesis were verified in tumor-bearing chickens infected with ALV-J and REV.

## 5. Conclusions

In conclusion, the current study revealed a synergistic oncogenicity mechanism induced by two simple retroviruses, ALV-J and REV. ALV-J and REV synergistically suppressed cellular senescence and activated EMT in vitro, and synergistically induced tumorigenesis and tumor spectrum extension in vivo. Mechanistically, as shown in Figure 10, after co-infection with ALV-J and REV, the released or expressed structural proteins from ALV-J and REV in the cytoplasm synergistically activated MSI1 expression, which directly targeted pri-miR-147 through its RNA binding site, causing the inhibition of miR-147 maturation. This relieved the inhibition of the NF-κB/KIAA1199/EGFR signaling pathway, thereby suppressing cellular senescence and activating EMT. The synergistic oncogenicity mechanism of ALV-J and REV sheds light on the identification of promising molecular targets and key barriers to the joint control of ALV-J and REV and the development of clinical technologies.

## Figures and Tables

**Figure 1 cells-11-03312-f001:**
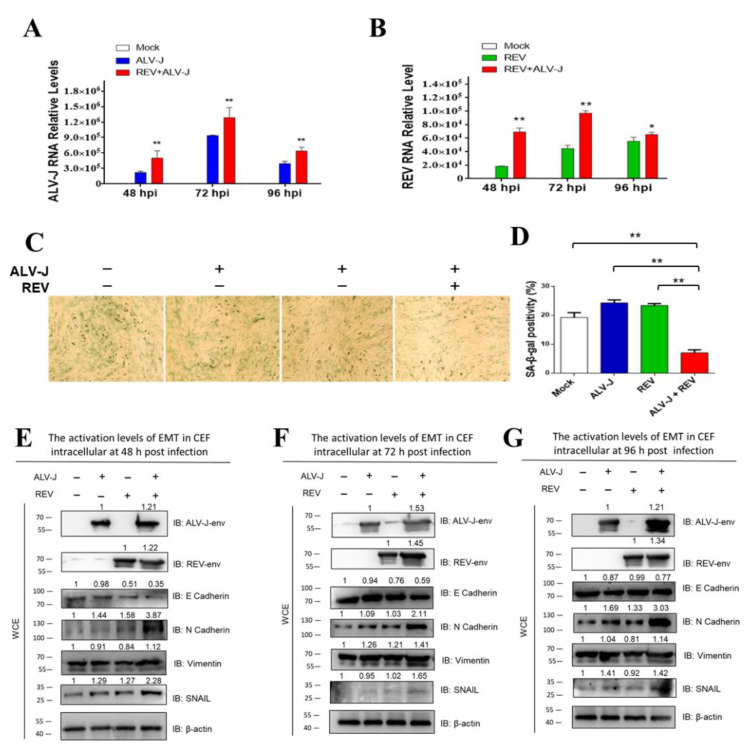
ALV-J and REV synergistically suppress cellular senescence and activate EMT in vitro. (**A**,**B**) ALV-J and REV synergistically promoted mutual replication in chicken embryo fibroblast (CEF) cells at 48 hpi, 72 hpi, and 96 hpi. The data are presented as mean ± SEM, and were obtained from three independent experiments (*n* = 3); each experiment contained triplicates. (**C**) ALV-J and REV synergistically inhibited CEF cellular senescence. CEFs were infected with either ALV-J, REV, or both, and examined via senescence-associated (SA) β-Gal staining on day 6 pi (100×). (**D**) Quantification of results shown in (**C**). ALV-J and REV synergistically activated EMT in CEF cells at 48 hpi (**E**), 72 hpi (**F**), and 96 hpi (**G**). ALV-J and REV synergistically enhanced N-cadherin, vimentin, and SNAIL protein levels, and synergistically decreased the E-cadherin protein level in CEF cells at 48 hpi, 72 hpi, and 96 hpi, as detected using WB. ** *p* ≤ 0.01 determined using Student′s *t*-test versus the Neg. Ctrl. group. * *p* ≤ 0.05 determined using Student′s *t*-test versus the Neg. Ctrl. group.

**Figure 2 cells-11-03312-f002:**
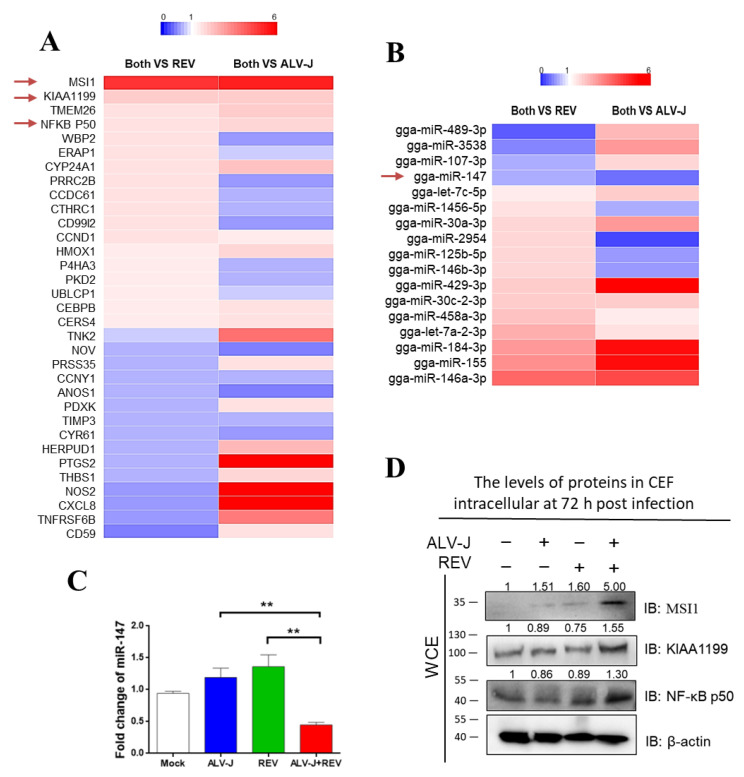
Identification of host molecules responsible for synergistic oncogenicity induced by ALV-J and REV. (**A**) Thirty-three differentially expressed proteins were common between cells co-infected with both viruses and infected with either ALV-J or REV. (**B**) miRNA whole-genome sequencing of infected CEF cells at 72 hpi. Seventeen differentially expressed miRNAs were common between co-infected and mono-infected cells. (**C**) ALV-J and REV infections synergistically inhibited the miR-147 expression level in CEF cells at 72 hpi, as detected using qPCR. (**D**) ALV-J and REV infections synergistically elevated MSI1, KIAA1199, and NF-κB p50 protein levels in DF-1 cells at 72 hpi, as detected using WB. ** *p* ≤ 0.01 determined using Student′s *t*-test versus the Neg. Ctrl. group.

**Figure 3 cells-11-03312-f003:**
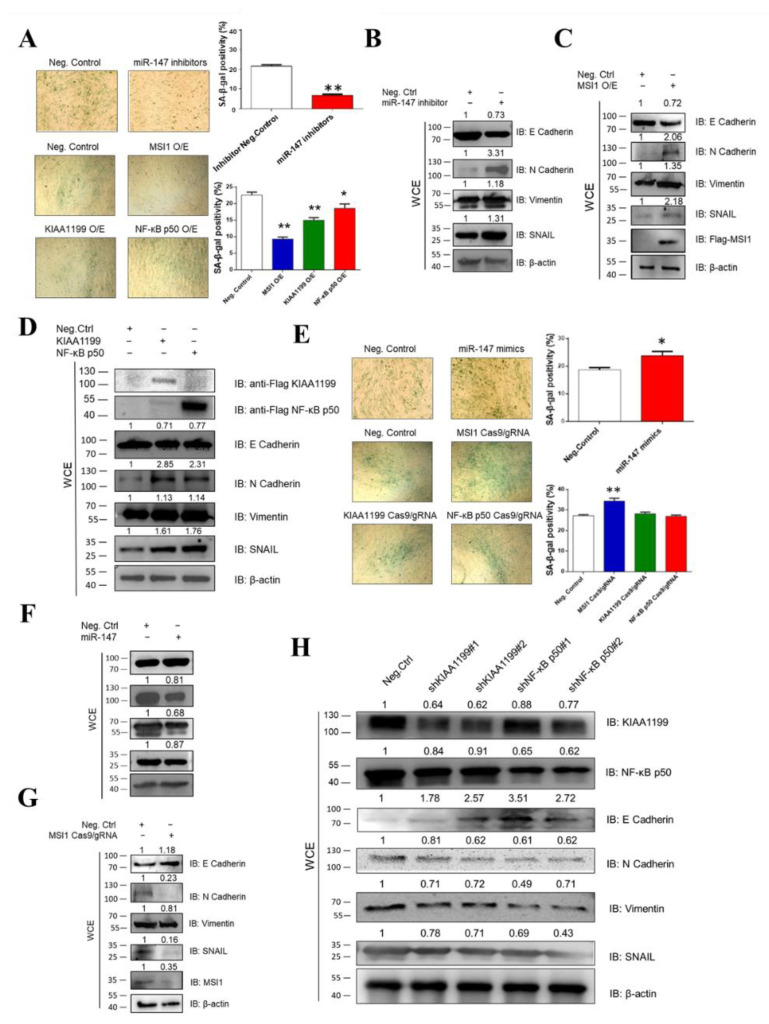
Overexpression of MSI1, KIAA1199, and NF-κB p50 and inhibition of miR-147 were all associated with oncogenicity. (**A**) Senescence-associated (SA) β-Gal staining revealed that miR-147 inhibitor, overexpression of MSI1, KIAA1199, and NF-κB p50 blocked cellular senescence. Negative control was transfected into CEFs at 4 dpi, after which SA β-Gal staining was performed at 48 h post-transfection (100×). (**B**) miR-147 inhibitor activated EMT in CEFs at 48 hpi, as detected using WB. (**C**) Overexpressing MSI1 activated EMT in CEFs at 48 hpi, as detected using WB. (**D**) Overexpressing KIAA1199 and NF-κB p50 activated EMT in CEFs at 48 hpi, as detected using WB. (**E**) miR-147 mimics and the inhibition of MSI1, KIAA1199, and NF-κB p50 promoted cellular senescence. Negative control was transfected into CEF cells at 4 dpi and SA β-Gal staining was performed 48 h post-transfection (100×). (**F**) miR-147 mimics decreased EMT in CEFs at 48 hpi, as detected using WB. (**G**) Inhibiting MSI1 decreased EMT of CEFs at 48 hpi, as detected using WB. (**H**) Inhibiting KIAA1199 and NF-κB p50 decreased EMT of CEFs at 48 hpi, as detected using WB. ** *p* ≤ 0.01 determined using Student′s *t*-test versus the Neg. Ctrl. group. * *p* ≤ 0.05 determined using Student′s *t*-test versus the Neg. Ctrl. group.

**Figure 4 cells-11-03312-f004:**
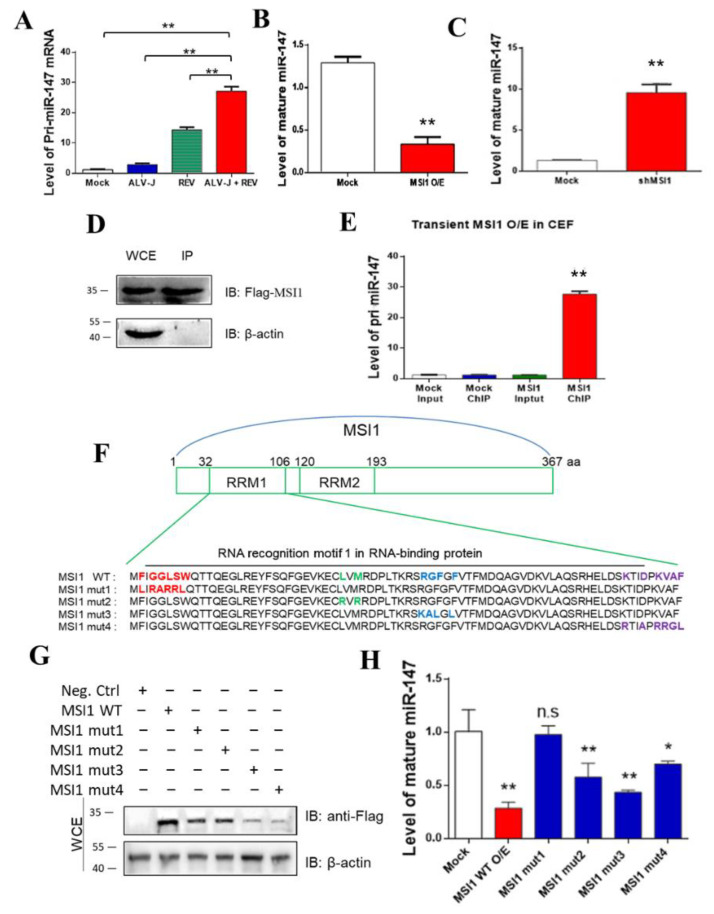
MSI1 directly targeted pri-miR-147 to inhibit miR-147 maturation. (**A**) ALV-J and REV synergistically elevated mRNA expression levels of pri-miR-147 in DF-1 cells, as detected using qPCR at 72 hpi. (**B**) mRNA expression level of mature miR-147 was suppressed by transfecting FLAG-MSI1 plasmid, as detected using qPCR at 48 hpi. (**C**) mRNA expression levels of mature miR-147 were increased by transfecting MSI1 Cas9/gRNA plasmid, as detected using qPCR at 48 hpi. (**D**,**E**) Accumulation of pri-miR-147 due to transient MSI1 expression in DF-1 cells, as detected using qPCR. Relative MSI1 expression levels were detected using WB with anti-FLAG antibody. RNA associated with immunopurified FLAG-MSI1 from DF-1 cells was analyzed using ChIP. RNA was extracted from IP material and analyzed using qPCR. (**F**) Strategy for constructing MSI1 mutants with different RRM1 regions. Original backbone of MSI1 is shown. The MSI1 mutants, including MS1 mut1 (F33L, G35R, G36A, L37R, S38R, and W39L), MS1 mut2 (L60R and M62R), MSI1 mut3 (R71K, G72A, F73L, and F75L), and MSI1 mut4 (K98R, D101A, K103R, V104R, A105G, and F106L) were constructed using the Fast Site-Directed Mutagenesis Kit. Mutations of amino acid sequences are shown in the same colors. (**G**) All MSI mutants were successfully transfected into DF-1 cells co-infected with ALV-J and REV, as detected using WB with anti-FLAG antibody at 48 hpi. (**H**) Among the four mutants, only MSI1 mut1 abolished the inhibition of miR-147 maturation, as detected by qPCR at 48 hpi. Data are presented as mean ± SEM for *n* = 3, with each experiment being performed in triplicate. ** *p* ≤ 0.01 determined using Student′s *t*-test versus the Neg. Ctrl. group. * *p* ≤ 0.05 determined using Student′s *t*-test versus the Neg. Ctrl. group. n.s., not significant.

**Figure 5 cells-11-03312-f005:**
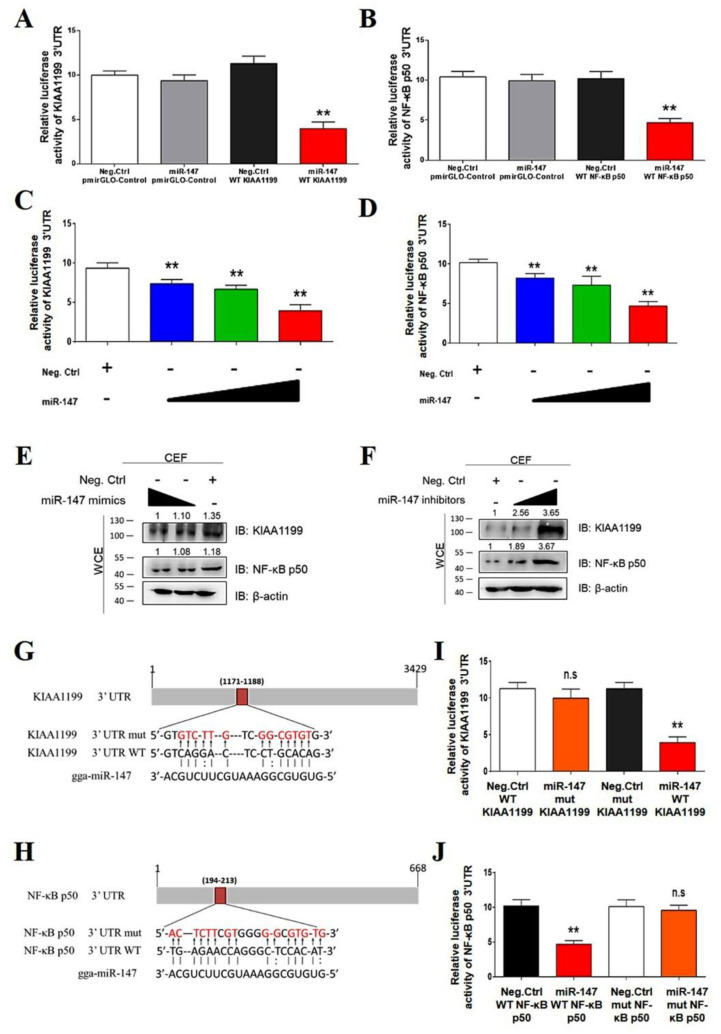
MiR-147 regulated NF-κB p50 and KIAA1199 by targeting the 3′ UTR. (**A**) miR-147 inhibited reporter activities of the pmirGLO-KIAA1199 3′ UTR. miR-147 mimics (40 nM) or negative controls were co-transfected with pmirGLO-Control or pmirGLO-KIAA1199 3′ UTR reporter plasmid into 293T cells. (**B**) miR-147 inhibited reporter activities of pmirGLO-NF-κB p50 3′ UTR. miR-147 mimics (40 nM) or negative controls with pmirGLO-Control or pmirGLO-NF-κB p50 3′ UTR reporter plasmid were co-transfected into 293T cells. (**C**) miR-147 mimics (10, 20, and 40 nM) or a negative control were co-transfected along with pmir-GL0-KIAA1199 3′ UTR reporter plasmid into 293T cells. (**D**) miR-147 mimics (10, 20, and 40 nM) or a negative control were co-transfected along with pmir-GL0- NF-κB p50 3′ UTR reporter plasmid into 293T cells. (**E**) miR-147 inhibited the expressions of KIAA1199 and NF-κB p50 in a dose-dependent manner. miR-147 mimics (20 and 40 nM) were transfected into ALV-J and REV co-infected CEF cells, after which WB was performed with anti-KIAA1199 antibody or anti- NF-κB p50 antibodies 48 h post-transfection. (**F**) Inhibition of miR-147 promoted the expression levels of KIAA1199 and NF-κB p50. miR-147 inhibitors (30 and 60 nM) were transfected into CEF cells, after which WB was performed with anti-KIAA1199 or anti- NF-κB p50 antibodies 48 h post-transfection. (**G**) Schematic of predicted seed sequence of miR-147 which binds with KIAA1199 3′ UTR. (**H**) Schematic of predicted seed sequence of miR-147 which binds with the NF-κB p50 3′ UTR. (**I**) KIAA1199 3′ UTR wild type (WT KIAA1199) was co-transfected with a negative control (Neg. Ctrl.) or miR-147 into 293T cells, whereas mutant KIAA1199 3′ UTR construct (mut KIAA1199) was co-transfected with Neg. Ctrl. or miR-147. (**J**) NF-κB p50 3′ UTR wild type (WT NF-κB p50) was co-transfected with a negative control (Neg. Ctrl.) or miR-147 into 293T cells, whereas mutant NF-κB p50 3′ UTR construct (mut NF-κB p50) was co-transfected with Neg. Ctrl. or miR-147. The above luciferase assays were performed 48 h later; data are presented as the mean ± SEM for *n* = 3, with each experiment being performed in triplicate. ** *p* ≤ 0.01 determined using Student′s *t*-test versus the Neg. Ctrl. group. n.s., not significant.

**Figure 6 cells-11-03312-f006:**
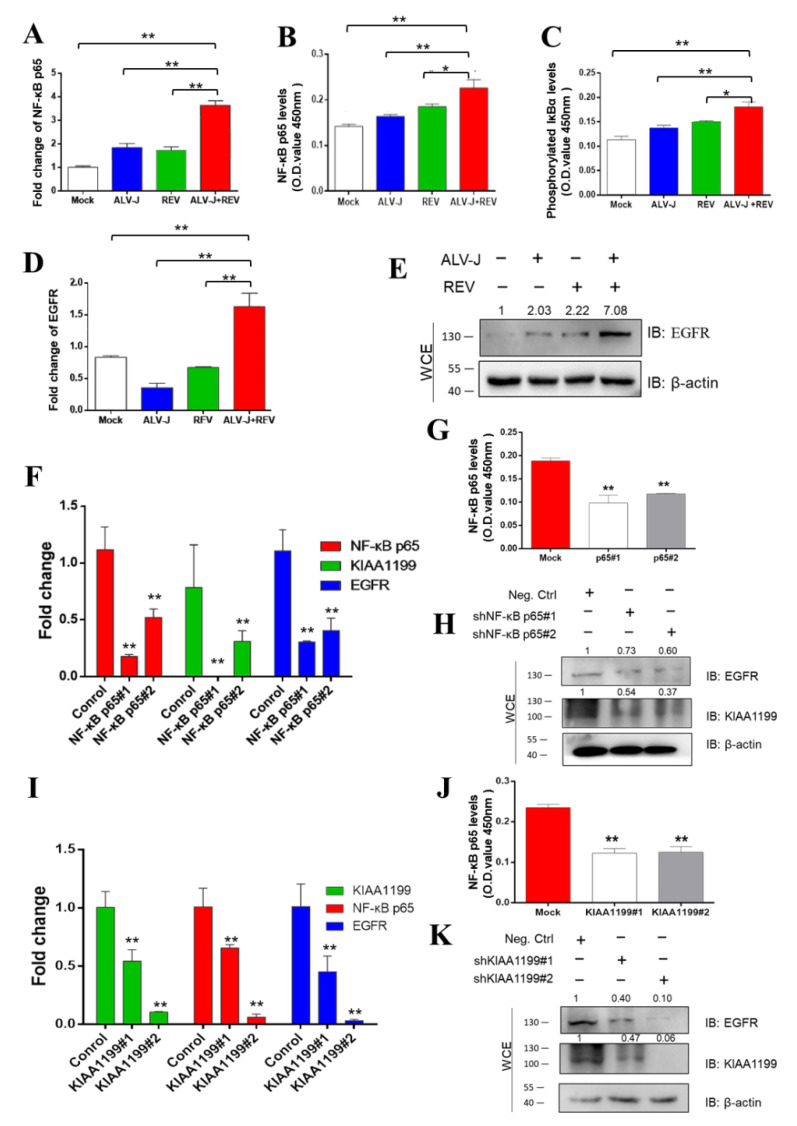
ALV-J and REV synergistically activated NF-κB/KIAA1199/EGFR signaling pathway crosstalk. (**A**) Co-infection with ALV-J and REV enhanced the NF-κB p65 mRNA expression levels in DF-1 cells at 72 hpi. (**B**) ALV-J and REV synergistically enhanced NF-κB p65 protein levels in DF-1 cells at 72 hpi, as detected using the NF-κB p65 ELISA kit. (**C**) ALV-J and REV synergistically enhanced phosphorylated IκBα expression levels in DF-1 cells at 72 hpi, as detected using the phosphorylated IκBα ELISA kit. ALV-J and REV synergistically enhanced EGFR mRNA expression (**D**) and protein (**E**) levels in DF-1 cells at 72 hpi. Protein levels were estimated using WB with anti-EGFR antibody. (**F**) mRNA expression levels of NF-κB p65, KIAA1199, and EGFR were suppressed after incubation with NF-κB p65 shRNA at 48 hpi. (**G**) NF-κB p65 protein levels were inhibited by NF-κB p65 shRNA, as detected using the NF-κB p65 ELISA kit. (**H**) Protein levels of KIAA1199 and EGFR were suppressed by NF-κB p65 shRNA at 48 hpi, as detected using WB with anti-KIAA1199 antibody and anti-EGFR antibody. (**I**) mRNA expression levels of KIAA1199, NF-κB p65, and EGFR were suppressed by KIAA1199 shRNA. (**J**) Protein levels of NF-κB p65 were inhibited by KIAA1199 shRNA, as detected using the NF-κB p65 ELISA kit. (**K**) Protein levels of KIAA1199 and EGFR were suppressed by KIAA1199 shRNA at 48 hpi, as detected using WB with anti-KIAA1199 and anti-EGFR antibodies. Data are presented as mean ± SEM for *n* = 3, with each experiment being performed in triplicate. ** *p* ≤ 0.01 determined using Student′s *t*-test versus the Neg. Ctrl. group. * *p* ≤ 0.05 determined using Student′s *t*-test versus the Neg. Ctrl. group.

**Figure 7 cells-11-03312-f007:**
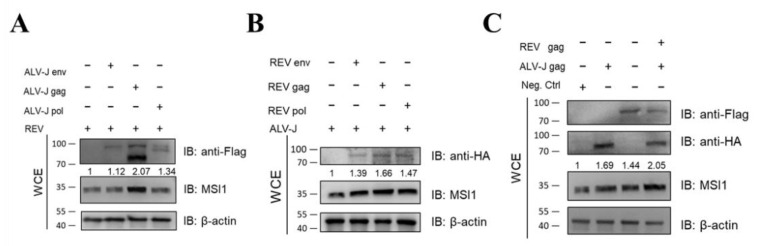
Structural proteins from ALV-J and REV synergistically activate MSI1. (**A**) ALV-J env, gag, and pol activated MSI1 protein expression level in DF-1 infected REV at 48 hpi, as detected using WB. Among the three viral structural proteins, ALV-J gag showed the most significant promotion effect on MSI1 protein expression. (**B**) REV env, gag, and pol activated MSI1 protein expression level in DF-1 infected ALV-J at 48 hpi, as detected using WB. Among the three viral structural proteins, REV gag showed the most significant promotion effect on MSI1 protein expression. (**C**) ALV-J gag and REV gag synergistically activated MSI1 protein expression level in CEF cells at 48 hpi, as detected using WB.

**Figure 8 cells-11-03312-f008:**
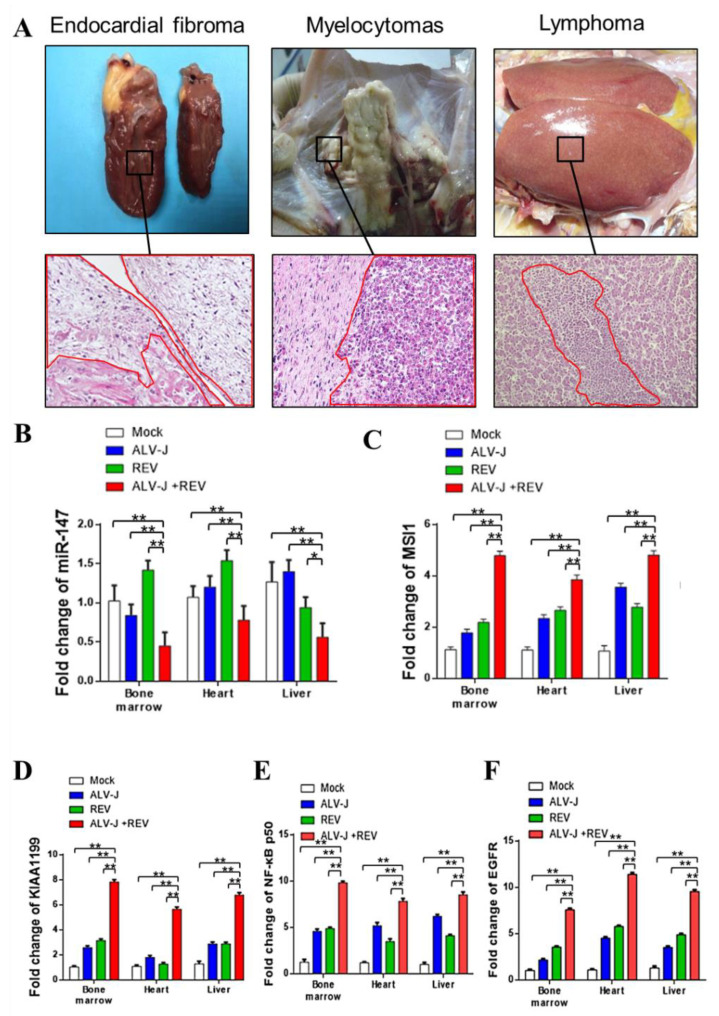
ALV-J and REV synergistically induced an increase in MSI1, KIAA1199, NF-κB p50, and EGFR expression and decreased miR-147 expression levels in vivo. (**A**) ALV-J and REV synergistically induced endocardial fibroma, myelocytomas, and lymphomas in the hearts, bone marrow, and livers of chickens, inserts at 400× magnification. Compared to the mono-infection group, RNA levels of MSI1 (**B**), KIAA1199 (**C**), NF-κB p50 (**D**), and EGFR (**E**) were significantly upregulated, and RNA levels of miR-147 (**F**) were significantly downregulated in the bone marrow, livers, and hearts of chickens in coinfection group. ** *p* ≤ 0.01 determined using Student′s *t*-test versus the Neg. Ctrl. group. * *p* ≤ 0.05 determined using Student′s *t*-test versus the Neg. Ctrl. group.

**Figure 9 cells-11-03312-f009:**
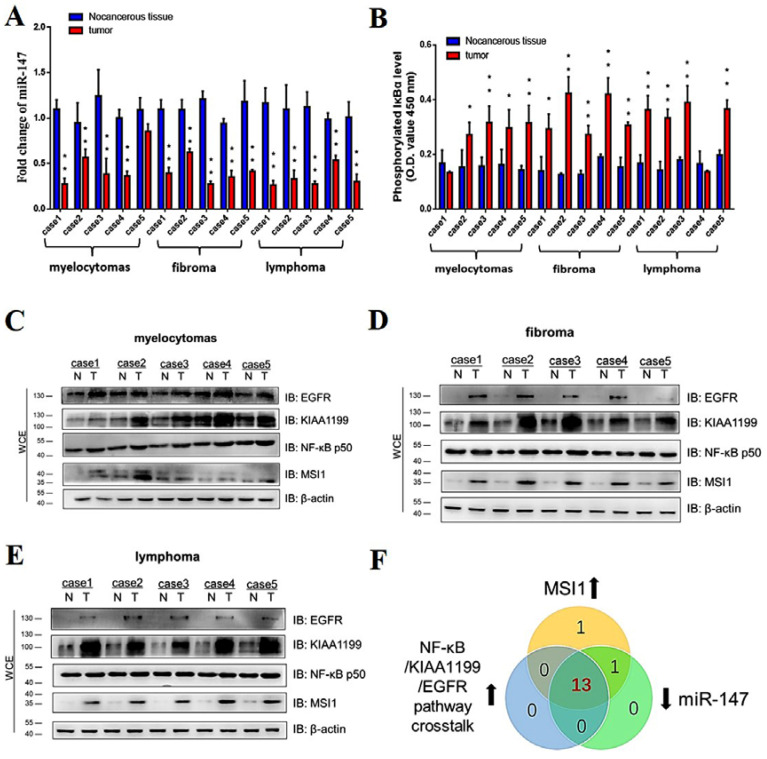
miR-147 was downregulated in ALV-J and REV coinfection-induced tumors and was negatively correlated with MSI1 and NF-κB/KIAA1199/EGFR pathway crosstalk. (**A**) Expression levels of miR-147 in 14 of 15 tumors were lower than those of corresponding non-cancerous tissues, as detected using qPCR. Data are presented as mean ± SEM for *n* = 3, with each experiment being performed in triplicate. (**B**) Phosphorylated IκBα expression levels in 13 of 15 tumors were elevated when compared to corresponding non-cancerous tissues, as detected using an NF-κB P-IκBα ELISA kit. Data are presented as mean ± SEM for *n* = 3, with each experiment being performed in triplicate. (**C**) Expression levels of EGFR, KIAA1199, NF-κB p50, and MSI1 in myelocytomas were detected using WB with anti-EGFR, anti-KIAA1199, anti-NF-κB p50, and anti-MSI antibodies, respectively. (**D**) Expression levels of EGFR, KIAA1199, NF-κB p50, and MSI1 in fibromas were detected using WB with anti-EGFR, anti-KIAA1199, anti-NF-κB p50, and anti-MSI antibodies, respectively. (**E**) Expression levels of EGFR, KIAA1199, NF-κB p50, and MSI1 in lymphomas were detected using WB with anti-EGFR, anti-KIAA1199, anti-NF-κB p50, and anti-MSI antibodies, respectively. (**F**) Comparisons of MIS1 elevation, miR-147 elevation, and NF-κB/KIAA1199/EGFR pathway crosstalk elevation in tumors. Central overlapping region represents all cases of MIS1, miR-147, and NF-κB/KIAA1199/EGFR pathway crosstalk elevations. ** *p* ≤ 0.01 determined using Student′s *t*-test versus the Neg. Ctrl. group. * *p* ≤ 0.05 determined using Student′s *t*-test versus the Neg. Ctrl. group.

**Figure 10 cells-11-03312-f010:**
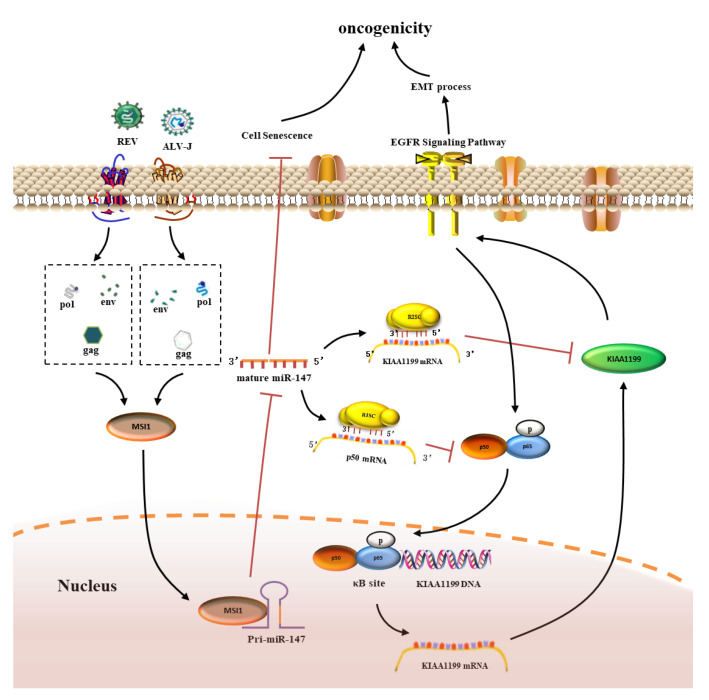
Schematic of the molecular mechanisms of synergistic oncogenicity induced by ALV-J and REV. After co-infection of ALV-J and REV, released or expressed structural proteins from ALV-J and REV in the cytoplasm synergistically elevate the expression levels of MSI1, which directly targets pri-miR-147, causing the inhibition of miR-147 maturation, which relieves the inhibition of the NF-κB/KIAA1199/EGFR signaling pathway and thereby suppresses cellular senescence and activates the EMT process, promoting oncogenicity.

## Data Availability

The mass spectrometry proteomics data have been deposited into the ProteomeXchange Consortium (http://proteomecentral.proteomexchange.org, http://pridb.gdcb.iastate.edu/RPISeq/, accessed on 28 September 2022) via the iProX partner repository with the dataset identifier PXD031503. The Illumina small-RNA deep sequencing data for the reported miRNAs have been deposited with the NCBI GEO under accession number GSE109105. All study data are included in the article and/or Appendix A.

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
