# Peer review of "Musashi-1 and miR-147 Precursor Interaction Mediates Synergistic Oncogenicity Induced by Co-Infection of Two Avian Retroviruses"

_cells, 2022, doi:10.3390/cells11203312_

Round 1

Reviewer 1 Report (Previous Reviewer 1)

I understand that Musashi-1 and miR-147 precursor interaction mediates synergistic oncogenicity by the co-infection. However, the manuscript has one problem. As I pointed out previously, the increased viral replication may be the major cause of the synergistic oncogenicity. For example, Musashi-1 expression was increased by each ALV infection along (Fig.2). This result suggests that the increased viral replication may enhance the extent of the up-regulation in the Musashi-1 expression. Thus, I would like to know how these viruses replicate more efficiently at the co-infection. Evidence showing that elevated expression of Musashi-1 enhances the viral replication or Musashi-1 knockdown inhibits the viral replication is very helpful.

Author Response

Thank you for your comments concerning our manuscript entitled " Musashi-1 and miR-147 precursor interaction mediates synergistic oncogenicity induced by co-infection of two avian retroviruses " (cells-1969996). The comment is valuable and the important guiding significance to our researches. We have studied the comment carefully and have made the answer which we hope meet with approval. The respond to the reviewer’s comment is listed as following:

Comment: I understand that Musashi-1 and miR-147 precursor interaction mediates synergistic oncogenicity by the co-infection. However, the manuscript has one problem. As I pointed out previously, the increased viral replication may be the major cause of the synergistic oncogenicity. For example, Musashi-1 expression was increased by each ALV infection along (Fig.2). This result suggests that the increased viral replication may enhance the extent of the up-regulation in the Musashi-1 expression. Thus, I would like to know how these viruses replicate more efficiently at the co-infection. Evidence showing that elevated expression of Musashi-1 enhances the viral replication or Musashi-1 knockdown inhibits the viral replication is very helpful.

Answer: Thank you for your valuable comments. We agree with the reviewer that increased viral replication may be a major cause of synergistic oncogenicity. In this study, the increase in MSI1 expression was indeed accompanied by the increase in viral replication. In fact, we first done the study of MSI1 related to viral replication of ALV-J and REV. However, although results of qRT-PCR showed MSI1 tended to promote the replication of ALV-J and REV, it was not significant (the relative figure 1 are shown below), indicating that the mechanism of MSI1 promoting synergistic oncogenesis was not fully dependent on the up-regulation of viral replication. That is the reason why we subsequently wanted to search for mechanisms of MSI1-mediated viral synergistic oncogenicity through host genes. We are sorry again for not describing the results of previous studies.

We also agree that enhanced viral replication may activate proto-oncogenes by insertional mutagenesis or by viral direct effects. In this study, we found all structural proteins of ALV-J and REV were involved in the synergistic activation of MSI1. In future studies, we would like to focus on the effect of viral integration mutations in the activation of MSI1 and the relationship between the increased viral replication and the synergistic oncogenicity.

Reviewer 2 Report (Previous Reviewer 2)

Accept.

Author Response

Thank you for your comments concerning our manuscript entitled " Musashi-1 and miR-147 precursor interaction mediates synergistic oncogenicity induced by co-infection of two avian retroviruses " (cells-1969996). The comments are valuable and the important guiding significance to our researches. 

Reviewer 3 Report (Previous Reviewer 3)

All the question are answered as required.

Author Response

Thank you for your comments concerning our manuscript entitled " Musashi-1 and miR-147 precursor interaction mediates synergistic oncogenicity induced by co-infection of two avian retroviruses " (cells-1969996). The comments are valuable and the important guiding significance to our researches. 

Round 2

Reviewer 1 Report (Previous Reviewer 1)

Now I understand the manuscript.

This manuscript is a resubmission of an earlier submission. The following is a list of the peer review reports and author responses from that submission.

Round 1

Reviewer 1 Report

This manuscript shows that some non-coding RNA molecules are associated with the tumorigenesis induced by co-infection of two different slowly transforming avian retroviruses. This study has a critical problem. It is generally thought that tumors in the retrovirus-infected animals are induced by insertional mutagenesis at a cellular oncogene or tumor suppressor gene through viral DNA integration as described in the Introduction section. There is no evidence showing that suppression of cellular senescence and activation of EMT in cultured cells represent the ALV-induced tumorigenesis in vivo. Are these biological events in cultured cells induced by the insertional mutagenesis? From this study I think that the co-infection activates the viral replication and enhance the opportunity of integration at cellular oncogenes or tumor suppressor genes. Is that the mechanism of synergistic tumorigenesis by the co-infection?

Author Response

Answer: Thank you for your valuable comments. We are sorry for not explaining why we chose to detect cellular senescence and EMT pathway activation as indicators of host cell tumorigenesis in vitro. Although both ALV-J and REV induce tumor formation in vivo, neither of them causes specific cytopathic effect in vitro. In order to adapt to synergistic infection with two retroviruses, host cells undoubtedly modify some biological traits. Therefore, we want to explore the mechanism of ALV-J and REV in tumorigenesis by using some indicators related to tumorigenesis in vitro, such as cellular senescence and Epithelial-Mesenchymal Transition (EMT). In this study, compared with the single infection group, co-infection of ALV-J and REV significantly suppress cellular senescence and activate EMT in vitro, which imply that inhibition of cell senescence and activation of EMT are essential pathways for the potential synergistic tumorigenesis of ALV-J and REV.

We fully agree with the reviewer that insertional mutagenesis is essential for retrovirus-induced tumorigenesis. Oncogenic retroviruses induce host proto-oncogene activation or tumor suppressor gene repression not only by insertional mutagenesis, but also by viral direct effects. Compared with single retroviral integration mutation, co-infection of two retroviruses may cause different types of co-integration mutations, including synergetic integration or antagonistic integration or no interference with each other. That is what we are going to focus on in future studies. In this study, we focused on the functions of key proteins and miRNAs of host cells associated with synergistic tumorigenesis induced by co-infection with two retroviruses, and found structural elements between two retroviruses were involved in the synergistic tumorigenesis process of ALV-J and REV. In further study, we will continue to explore the role of integration mutations in synergistic activation of MSI1 and even synergistic tumorigenesis induced by two retroviruses.

Reviewer 2 Report

The current study reveals a synergistic tumorigenesis mechanism induced by two simple retroviruses, ALV-J and REV. ALV-J and REV synergistically suppressed cellular senescence and activated EMT in vitro, and synergistically induced tumorigenesis and tumor spectrum extension in vivo. which directly targeted pri-miR-147 through its RNA binding site, causing miR-147 maturation inhibition. This relieved the inhibition of the NF-κB/KIAA1199/EGFR signaling pathway, thereby suppressing cellular senescence and activating EMT, promoting tumorigenesis.

Major issues:

The way this current manuscript is written, makes it difficult for a reader (or reviewer) to understand what has been done exactly. The intended message is cloaked by improper use of English grammar and writing style. The language must be improved.

The materials and methods section is the worst part of the manuscript. It lacks consistency and is impossible to follow – let alone if one wants to replicate the authors experiments.

It lacks a section on the cell lines used and their origin (human, chicken or whatever).

The authors mention ALV or REV proliferation, but only show viral RNA replication. In theory, viral RNA replication does not automatically lead to productive virions. Authors should include an assay to prove real virus propagation (like plaque assays or other experiments to prove viable virus output) if they want to make the statement that in co-infected cells significantly increased.

The authors demonstrate that co-infected induced NF-κB activation in Fig 2, 3, 5 and 9. Examination of nucleus translocation of NF-κB is suggested.

Minor issues:

In the abstract, for the first time show EMT, the full title should be added.

In the line of 72 to 73, what is mean of 103.8 50% TCID50? Why the quantitative value of TCID50 for REV and ALV is the same?

In the line of 108 and 109, the description of numbers is not uniform.

Author Response

Dear Reviewer:

Thank you for your comments concerning our manuscript entitled " Musashi-1 and miR-147 precursor interaction mediates synergistic tumorigenesis induced by co-infection of avian retroviruses in chicken" (cells-1905277). Those comments are all valuable and very helpful for revising and improving our paper, as well as the important guiding significance to our researches. We have studied comments carefully and have made correction which we hope meet with approval. Revised portion are marked in red in the paper. The main corrections in the paper and the responds to the reviewer’s comments are listed as following:

Comment: The way this current manuscript is written, makes it difficult for a reader (or reviewer) to understand what has been done exactly. The intended message is cloaked by improper use of English grammar and writing style. The language must be improved.

Answer: Thank you for your valuable comments. We have revised language throughout the manuscript by Elsevier Language Editing Services (Serial number: LE-233010-B30A62D1BEBE).

Comment: The materials and methods section is the worst part of the manuscript. It lacks consistency and is impossible to follow – let alone if one wants to replicate the authors experiments.

Answer: Thank you for your valuable comments. We have revised the language of materials and methods section.

Comment: It lacks a section on the cell lines used and their origin (human, chicken or whatever).

Answer: Thank you for your valuable comments. We have corrected it in the materials and methods section.

Comment: The authors mention ALV or REV proliferation, but only show viral RNA replication. In theory, viral RNA replication does not automatically lead to productive virions. Authors should include an assay to prove real virus propagation (like plaque assays or other experiments to prove viable virus output) if they want to make the statement that in co-infected cells significantly increased.

Answer: Thank you for your valuable comments. Neither ALV-J or REV causes specific cytopathic effect in vitro, so we have tested the expression levels of the key structural protein env of ALV-J and REV to further demonstrate virus replication. The results are shown in Figures 1E-G.

Comment: The authors demonstrate that co-infected induced NF-κB activation in Fig 2, 3, 5 and 9. Examination of nucleus translocation of NF-κB is suggested.

Answer: Thank you for your valuable comments. Considering the Reviewer’s suggestion, we have examined the nucleus translocation levels of NF-κB in normal CEF, CEF infected with single virus, and CEF co-infected with ALV-J and REV at 72 hpi by IFA assay, respectively. The results are shown in Figure S1. We have added it in line 364-366 of new manuscript.

Comment: In the abstract, for the first time show EMT, the full title should be added.

Answer: Thank you for your careful reading of our manuscript. We have corrected it in the abstract section.

Comment: In the line of 72 to 73, what is mean of 103.8 50% TCID50? Why the quantitative value of TCID50 for REV and ALV is the same?

Answer: Thank you for your careful reading of our manuscript. They are unfortunate clerical errors. The stock SNV strain of REV at 103.2 TCID50 and the NX0101 of ALV-J 103.8 TCID50 were used in this study. We have corrected it in the materials and methods section.

Comment: In the line of 108 and 109, the description of numbers is not uniform.

Answer: Thank you for your careful reading of our manuscript. We have corrected it in the new manuscript.

Reviewer 3 Report

In this study the authors suggest a synergistic tumorigenesis mechanism induced by two simple avian oncogenic retroviruses, ALV-J and REV. They make the conclusion that two retroviruses synergistically activate MSI1 expression, which directly targeted pri-miR-147 through its RNA binding site, causing miR-147 maturation inhibition. This relieved the inhibition of the NF-κB/KIAA1199/EGFR signaling pathway, thereby suppressing cellular senescence, activating EMT and promoting tumorigenesis. The data presented by this study have contributed to elucidate tumorigenic mechanism of ALV-J and REV co-infection, and even the develop tumor control strategies for humans and other animals. There are some questions need to be addressed before accept to publish.

Comments:

1.      Why did the authors choose to detect the cellular senescence to refer to indicators of cell tumorigenesis in vitro? It needs more explanation in the manuscript.

2.      Are data from CEF in vitro relevant to pathologies in vivo by ALV-J and REV since they infect different cell types, such as B-lymphocytes and endothelial cells, rather than fibroblasts?

3.      To understand whether two retroviruses synergistically affected tumorigenesis, EMT were measured in ALV-J and REV co-infected cells at 48, 72 and 96 hpi. The reason why these time points are selected for the test needs to be explained.

4.      CEFs co-infected with ALV-J and REV, mono-infected with ALV-J or REV, and Mocks were analyzed using proteomics combined with miRNA whole-genome sequencing analysis at 72 hpi. Why choose 72 hours as the time point? Need to explain.

5.      Some Figures are too small to be readable with poor resolution owing to being scanned in. Therefore the reader cannot discern the numbers on the Y-axis, such as Figure 3A, 3E, 4B, 4C and 6C.

6.      There are some grammatical errors in the manuscript. It needs to ask a native English speaker to further polish the language.

Author Response

Dear Reviewer:

Thank you for your comments concerning our manuscript entitled " Musashi-1 and miR-147 precursor interaction mediates synergistic tumorigenesis induced by co-infection of avian retroviruses in chicken" (cells-1905277). Those comments are all valuable and very helpful for revising and improving our paper, as well as the important guiding significance to our researches. We have studied comments carefully and have made correction which we hope meet with approval. Revised portion are marked in red in the paper. The main corrections in the paper and the responds to the reviewer’s comments are listed as following:

1) Comment:   Why did the authors choose to detect the cellular senescence to refer to indicators of cell tumorigenesis in vitro? It needs more explanation in the manuscript.

Answer: Thank you for your valuable comments. We are sorry for not explaining why we chose to detect cellular senescence as indicators of host cell tumorigenesis in vitro. Although both ALV-J and REV induce tumor formation in vivo, both of them did not cause specific cytopathic effect in vitro. In order to adapt to synergistic infection with two retroviruses, host cells undoubtedly modify some biological traits. Therefore, we want to explore the mechanism of ALV-J and REV in tumorigenesis by using some indicators related to tumorigenesis in vitro, such as cellular senescence. In this study, compared with the single infection group, co-infection of ALV-J and REV significantly suppress cellular senescence in vitro, which implies that inhibition of cell senescence is essential pathway for the potential synergistic tumorigenesis of ALV-J and REV.

2) Comment: Are data from CEF in vitro relevant to pathologies in vivo by ALV-J and REV since they infect different cell types, such as B-lymphocytes and endothelial cells, rather than fibroblasts?

Answer: Yes, the data showed the released or expressed structural proteins from ALV-J and REV in the cytoplasm synergistically activated MSI1 expression, which directly targeted pri-miR-147 through its RNA binding site, causing miR-147 maturation inhibition, activating the NF-κB/KIAA1199/EGFR signaling pathway, and thereby suppressing cellular senescence and activating EMT in CEF cells. CEF, as a normal cell model for studying REV and ALV-J, abound in the many organs of the chicken, which may reflect more common synergistic mechanism than B-lymphocytes or endothelial cells. In further study, we will choose the B-lymphocytes as the cell model to explore immunosuppression mechanism of co-infection ALV-J with REV.

3) Comment:   To understand whether two retroviruses synergistically affected tumorigenesis, EMT were measured in ALV-J and REV co-infected cells at 48, 72 and 96 hpi. The reason why these time points are selected for the test needs to be explained.

Answer: Thank you for your valuable comments. We are also sorry for not explaining the study design and convince the audience that all the discoveries of this model come from synergistic infection. Our previous studies showed both ALV-J and REV levels in the co-infection group were increased significantly compared to those in the single infection groups at 48 hpi, 72 hpi, 96 hpi, 120 hpi and 144 hpi and reached the highest peak at 72 hpi [1]. In this study, we wanted to explore the synergistic tumorigenesis induced by co-infection with ALV-J and REV at the before and after the time point when viral replication reached the highest peak.

4) Comment: CEFs co-infected with ALV-J and REV, mono-infected with ALV-J or REV, and Mocks were analyzed using proteomics combined with miRNA whole-genome sequencing analysis at 72 hpi. Why choose 72 hours as the time point? Need to explain.

Answer: Our previous studies showed that REV and ALV-J synergistically increased the accumulation of exosomal miRNAs at 72 hpi [1], so we also performed this study at 72 hpi to explore the synergistic mechanisms of REV and  ALV-J at the miRNA level and proteome level in CEF cells.

5) Comment: Some Figures are too small to be readable with poor resolution owing to being scanned in. Therefore the reader cannot discern the numbers on the Y-axis, such as Figure 3A, 3E, 4B, 4C and 6C.

Answer: Thank you for your careful reading of our manuscript. We have corrected it in the new manuscript.

6) Comment: There are some grammatical errors in the manuscript. It needs to ask a native English speaker to further polish the language.

Answer: Thank you for your careful reading of our manuscript. We have corrected it in the new manuscript.

References

  1. Zhou D, Xue J, He S, Du X, Jing Z, Li C, Huang L, Nair V, Yao Y, Cheng Z: Reticuloendotheliosis virus and avian leukosis virus subgroup J synergistically increase the accumulation of exosomal miRNAs. Retrovirology 2018, 15(1):45.

Round 2

Reviewer 1 Report

I understand that Musashi-1 and miR-147 are associated with the suppression of cellular senescence and activation of EMT by the co-infection in cultured cells. However, you cannot say that Musashi-1 and miR-147 are associated with "the synergistic tumorigenesis".  Tumorigenesis means tumor formation. Did the co-infection induce tumors in cultured cells? You answered that you want to explore the mechanism of ALV-J and REV in tumorigenesis by using some indicators related to tumorigenesis in vitro, such as cellular senescence and Epithelial-Mesenchymal Transition (EMT).There is no evidence showing that suppression of cellular senescence and activation of EMT in cultured cells represent tumorigenesis. Although the co-infection increased the expression of MSI1, KIAA1199, NF-kB p50, and EGFR in vivo (Fig.8), It is possible that the increaset is induced after tumorigenesis. Evidence showing that the increased expression of these factors in vivo is pre-requisite for tumorigenesis is absolutely necessary. For example, the co-infection does not induce synergistic tumorigenesis in Musashi-1-deficient animals. I know that experiment is very difficult, but from these results you can conclude that Musashi-1 and miR-147 are associated with the suppression of cellular senescence and activation of EMT by the co-infection in cultured cells, but not tumorigenesis.

Reviewer 2 Report

Accept.